# A Comparative Analysis of the Physico-Chemical Properties of Pectin Isolated from the Peels of Seven Different Citrus Fruits

**DOI:** 10.3390/gels9110908

**Published:** 2023-11-16

**Authors:** Khodidash Baraiya, Virendra Kumar Yadav, Nisha Choudhary, Daoud Ali, Daya Raiyani, Vibhakar A. Chowdhary, Sheena Alooparampil, Rohan V. Pandya, Dipak Kumar Sahoo, Ashish Patel, Jigna G. Tank

**Affiliations:** 1Department of Biosciences, Saurashtra University, Rajkot 360005, India; baraiyak828@gmail.com (K.B.); diyaraiyani1256@gmail.com (D.R.); vibhakarchowdhary@gmail.com (V.A.C.); jforsree@gmail.com (S.A.); 2Department of Life Sciences, Hemchandracharya North Gujarat University, Patan 384265, India; yadava94@gmail.com; 3Department of Zoology, College of Science, King Saud University, P.O. Box 2455, Riyadh 11451, Saudi Arabia; 4Department of Microbiology, Atmiya University, Rajkot 360005, India; rohan.pandya@atmiyauni.ac.in; 5Department of Veterinary Clinical Sciences, College of Veterinary Medicine, Iowa State University, Ames, IA 50011, USA; dsahoo@iastate.edu

**Keywords:** natural polymers, pectin, citrus fruits, hydrogel, anhydrouronic acid, galacturonic acid

## Abstract

In the present research work, pectin was isolated from the peels of seven citrus fruits (*Citrus limon*, *Citrus limetta*, *Citrus sinensis*, *Citrus maxima*, *Citrus jambhiri*, *Citrus sudachi*, and *Citrus hystrix*) for a comparison of its physicochemical parameters and its potential use as a thickening agent, gelling agent, and food ingredient in food industries. Among the seven citrus fruits, the maximum yield of pectin was observed from *Citrus sudachi,* and the minimum yield of pectin was observed from *Citrus maxima*. The quality of each pectin sample was compared by using parameters such as equivalent weight, anhydrouronic acid (AUA) content, methoxy content, and degree of esterification. It was observed that all seven pectin samples had a high value of equivalent weight (more than 1000), suggesting that all the pectin samples had a high content of non-esterified galacturonic acid in the molecular chains, which provides viscosity and water binding properties. The methoxy content and degree of esterification of all the pectins was lower than 50%, which suggests that it cannot easily disperse in water and can form gel only in presence of divalent cations. The AUA content of all isolated pectins samples was above 65%, which suggests that the pectin was pure and can be utilized as a food ingredient in domestic foods and food industries. From the FTIR analysis of pectin, it was observed that the bond pattern of *Citrus maxima*, *Citrus jambhiri*, and *Citrus hystrix* was similar. The bond pattern of *Citrus limon*, *Citrus limetta*, and *Citrus sinensis* was similar. However, the bond pattern of *Citrus sudachi* was different from that of all other *citrus* fruits. The difference in the bond pattern was due to the hydrophobic nature of pectin purified from *Citrus limon*, *Citrus limetta*, *Citrus sudachi*, and *Citrus sinensis* and the hydrophilic nature of pectin purified from *Citrus maxima*, *Citrus jambhiri*, and *Citrus hystrix.* Hence, hydrophobic pectin can be utilized in the preparation of hydrogels, nanofibers, food packaging material, polysoaps, drug delivery agents, and microparticulate materials, whereas hydrophilic pectin can be utilized for the preparation of gelling and thickening agents.

## 1. Introduction

Pectin is a multifaceted polysaccharide made up of homogalacturonan, which has residues of (1→4)-linked a-D-galacturonic acid; rhamnogalacturonan-I, which has (1→4)-linked a-D-galacturonic acid and (1→2)-linked a-L-rhamnose that are combined with each other alternatively through a backbone; rhamnogalacturonan II, having a backbone of (1→4)-linked α-D-galacturonic acid; and other unique sugars such as 3-deoxy-D-lyxo-2-heptulosaric acid (Dha), D-apiose, and 2-keto-3-deoxy-D-manno-2 octulosonic acid (Kdo) [1,2]. These are present in the side chains, which are attached to O-2 or O-3 of the galacturonic acid backbone. Some of the carboxyl, methyl-esterified, and hydroxyl groups, which are partially acetylated at O-2 or O-3, have intense effects on the functional properties of pectin [3,4]. However, some of the rhamnose residues contain arabinan, galactan, and arabinogalactan at 4-O-rhamnose in the side chains [5]. 

Pectin is a biodegradable, non-toxic, and water-soluble polysaccharide, which can form gels and has the binding capacity to form network structures [6]. It has significant applications in food industries as a thickener, gelling agent, stabilizer, emulsifier, and texturizer [7,8]. It is used worldwide in the production of desserts, jams, edible films, thick fruit juices, jellies, fruit drink essences, baking products, and dairy products [9,10]. It is also utilized in the preparation of adhesives, biodegradable films, materials for implantation in clinics, drug delivery, and plasticizers [11]. It has pharmaceutical activities such as antioxidation; inhibition of lipase activity; induction of apoptosis in human cancer cells; healing of wounds; stimulation of immune response; immunomodulation; maintenance of cholesterol levels; and antitussive, astringent, and anti-diabetic properties [12,13,14,15,16,17,18]. It decreases low-density lipoprotein cholesterol fractions and reduces the risk of coronary heart diseases [19,20]. Various dietary products made up of pectin fibers are used for the treatment of bowel cancer and hyperlipidemia [21]. Various previous studies have reported the presence of pectin in the peels of citrus fruits such as *Citrus limon*, *Citrus hystrix, Citrus tankan, Citrus maxima*, *Citrus limetta, Citrus sinensis* [22], and *Citrus depressa* [23,24,25,26,27,28,29].

Hence, the present study was carried out to screen wild citrus fruits as potential sources of good-quality pectin for potential use in the food and pharma industries. Wild citrus fruits such as *Citrus maxima*, *Citrus jambhiri*, *Citrus sudachi*, and *Citrus hystrix* are traditionally utilized for the preparation of vinegar and citric acid. However, their peels are a major biological waste [6] that is degraded without utilization and can be used commercially for the isolation of pectin. However, a comparative analysis is required to compare the qualitative and quantitative parameters of pectin isolated from common and wild citrus fruits for its possible use either as hydrogels, thickening agents, or emulsifiers in food and pharmaceutical industries.

## 2. Results and Discussion

In the present study, the purification of pectin from different citrus fruit peels was achieved through the mechanical grinding of peels in deionized water followed by alcohol precipitation at room temperature. Here, the fine grinding of peels up to nanoform is the crucial process in the extraction of pectin from citrus fruit peels. Extraction at room temperature and precipitation through the solvent method helps maintain the structural stability of the pectin without a loss of yield. 

### 2.1. Yield of Pectin

The yield of pectin obtained from *Citrus limon, Citrus limetta, Citrus sinensis, Citrus maxima, Citrus jambhiri, Citrus sudachi,* and *Citrus hystrix* was 37.89%, 34.24%*,* 30.61%, 11.68%, 12.91%, 54.29%, and 23.87%, respectively. Among all the citrus fruits, the highest pectin yield was obtained from *Citrus sudachi*, whereas the lowest pectin yield was obtained from *Citrus maxima* (Figure 1 and Figure 2). The variation in the yield of pectin depends on the water content and concentration of pectin in the peels. The yield of pectin obtained from *Citrus limon, Citrus limetta, Citrus sinensis, Citrus maxima, Citrus jambhiri, Citrus sudachi,* and *Citrus hystrix* was 37.89%, 34.24%, 30.61%, 11.68%, 12.91%, 54.29%, and 23.87%, respectively, which is almost similar to that in previous studies on pectin isolation from the peels of various fruits. 

A team led by Kamal isolated pectin from *Citrus sinensis* at different temperatures (65, 75, 85, and 95 °C)*,* pHs (1.0, 1.5, 2, and 2.5), and times (45, 60, 75, and 90 min) and suggested that the optimum condition to isolate pectin is 94.13 °C temperature, 1.45 pH, and a time of 114.70 min, which yields 23.64% pectin [26]. A team led by Devi isolated pectin from *Citrus limetta* at different pHs, temperatures, and times using citric acid and nitric acid. The investigator suggested that the highest pectin yields of 76.0% were observed for extraction in the presence of citric acid at a pH of 1.5, temperature of 80 °C, and a time of 60 min [30]. Then, they isolated pectin from *Citrus hystrix* using sun-dried and microwave-dried peels in the presence of citric acid, hydrochloric acid, or nitric acid at different temperatures (45, 65, and 90 °C) using a constant pH of 1.5 and a time of 1 h. They observed that the pectin yield in sun-dried peels ranged between 10.4% and 59.30% and that in microwave-dried peels ranged between 25.9% and 61.80%. Kanmani (2014) isolated pectin from citrus fruit peels and suggested that the pectin yield from *Citrus sinensis, Citrus limetta,* and *Citrus limon* was 29.41%, 32.42%, and 36.71%, respectively [31]. They suggested that the optimum conditions required for the extraction of pectin from these citrus fruits are heating at 65 °C temperature for 67.5 min at pH 3.5. Azad (2014) standardized pectin isolation from lemon pomace using distilled water, different acids (6 M HCl, 1 N H_2_SO_4_, 1 N HNO_3_, 6.2 g/100 g citric acid, 1 N acetic acid), and a combination of acetic acid and ammonium oxalate at different temperatures (70, 80, 90, and 100 °C), time intervals (30, 60, 90, and 120 min), and different stages of fruits. They suggested after analysis that lemon pomace made from premature fruit is the best pectin source with a yield ranging from 10.83% to 13.13% and that the optimum condition for the isolation of pectin is 100 °C for 60 min. Ahmed and their team isolated pectin from ginger lemon, cardamom lemon, and China lemon by using organic tartaric acid and 95% ethanol and obtained a yield of 5.71% of pectin from ginger lemon, 8.08% from cardamom lemon, and 12.73% from China lemon [32]. A team led by Methacanon isolated pectin from pomelo (*Citrus maxima*) and suggested that the optimum condition for the extraction of pectin from *Citrus maxima* is using nitric acid at a pH of 2 and temperature of 90 °C for 90 min, which gives a yield of pectin of up to 23.19% [24]. Dananjaya and their team isolated pectin from *Citrus maxima* and suggested that the yield of pectin was 14.5% [25]. A team led by Norziah isolated pectin from pomelo fruit peels and suggested that the yield of pectin was 20.8% [28]. Tamaki and Konisha (2008) isolated pectin from the endocarp of *Citrus depressa* and suggested that the yield of pectin was 4.1% [23]. Ghoshal and Negi (2020) isolated pectin from kinnow peels (*Citrus nobilis* + *Citrus deliciosa*) by hot extraction at 90 °C followed by ethanol precipitation and suggested that the yield of pectin was 6.13% [28]. Simpson and Morris (2014) observed that pectin from lemon peel, lime peel, apple pomace, and orange peel had pectin contents of 48%, 26%, 14%, and 11%, respectively [12].

### 2.2. Equivalent Weight

The equivalent weight is an important criterion for checking the quality characteristics of the isolated pectin. It is the total content of free galacturonic acid (not esterified) present in the molecular chains of pectin. The presence of free galacturonic acid in the molecular chains of pectin helps in providing viscosity and water-binding properties to the pectin. The equivalent weight of *Citrus limon, Citrus limetta, Citrus sinensis, Citrus maxima, Citrus jambhiri, Citrus sudachi,* and *Citrus hystrix* was 3077 g/mol, 7692 g/mol, 10,000 g/mol, 1250 g/mol, 5000 g/mol, 3333 g/mol, and 5000 g/mol, respectively. Among all the citrus fruits, the highest equivalent weight was observed in *Citrus sinensis,* whereas the lowest equivalent weight was observed in *Citrus maxima* (Figure 3). This suggests that a sufficient amount of free galacturonic acid is present in pectin isolated from all citrus fruits. Hence, they can be a good source of a thickening agent in the food [33,34] and cosmetic industries. 

The equivalent weight of pectin isolated from all the citrus fruits was high, which suggests that all the pectin samples are rich sources of free galacturonic acid, which provides viscosity, and has good water-binding properties that provides the gel-forming ability to pectin [28,35]. Similarly, Altaf and Egardt (2016) suggested that equivalent weight is the total content of free galacturonic acid in pectin [36]. A team led by Muthukumaran suggested that the equivalent weight of pectin varies on the basis of the pH and solvent used for extraction and the number of free acids available on it [37]. A group led by Wongkaew suggested that the higher equivalent weight reflects that pectin has a high gel-forming ability [38]. In previous studies, Kamal and their group suggested that the equivalent weight of pectin isolation from *Citrus sinensis* ranged between 1744.66 and 1899.33 g/mol [26]. Devi and their team suggested that the equivalent weight of pectin isolated from *Citrus limetta* ranged between 312.5 and 833.33 g/mol [30]. Kumar and their group suggested that the equivalent weight of pectin isolated from *Citrus hystrix* was 234.742 g/mol [29]. Kanmani (2014) suggested that the equivalent weight of pectin isolation from *Citrus sinensis, Citrus limetta,* and *Citrus limon* was 594.86 g/mol, 386.45 g/mol, and 253.70 g/mol, respectively [31]. Azad (2014) suggested that the equivalent weight of pectin isolated from lemon pomace ranged between 368 ± 3 and 1632 ± 137 g/mol [39]. A team led by Ahmed suggested that the equivalent weight of pectin isolated from three varieties of *Citrus limon* was 298 ± 21 g/mol in ginger lemon, 532 ± 24 g/mol in cardamom lemon, and 301 ± 12 g/mol in China lemon [32]. A team led by Dananjaya isolated pectin from *Citrus maxima* and suggested that the equivalent weight was 1245.65 g/mol [25]. A team led by Norziah isolated pectin from pomelo fruit peels and suggested that the equivalent weight was 391.2 g/mol [28].

### 2.3. Methoxyl Content

The methoxyl content of pectin determines its ability to disperse in water and form a hydrogel. The methoxyl content of pectin isolated from *Citrus limon, Citrus limetta, Citrus sinensis, Citrus maxima, Citrus jambhiri, Citrus sudachi,* and *Citrus hystrix* was 34.1%, 31%, 32.55%, 34.87%, 23.25%, 38.75%, and 38.75%, respectively. Among all the citrus fruits, the highest methoxyl content was observed in *Citrus sudachi* and *Citrus hystrix,* whereas the lowest methoxyl content was observed in *Citrus jambhiri* (Figure 4). All the pectin samples had a methoxy content lower than 50%; this suggests that they cannot easily disperse in water and are hydrophobic in nature.

In the current investigation, the methoxyl content of pectin isolated from the peels of all *Citrus* fruits was lower than 50%, which suggests that all are low/methoxy pectins that are hydrophobic in nature and will not disperse in water easily. Norziah and their team suggested that the methoxyl value signifies the distribution capacity of pectin in water. Its gel-forming capacity is due to a high-methoxyl content that determines strong adhesive and cohesive properties, which determine the augmentation of firmness in the food products [28]. In previous studies, Kamal and their group suggested that *Citrus sinensis* has low-methoxy pectin with a methoxyl content ranging between 5.02 and 5.64% [26]. A team led by Devi suggested that *Citrus limetta* has low-methoxy pectin with a methoxyl content ranging between 6.2% and 5.27% [30]. Kumar and their group suggested that *Citrus hystrix* has a low methoxy pectin with a methoxyl content of 11.86% [29]. Kanmani (2014) suggested that *Citrus sinensis*, *Citrus limetta*, and *Citrus limon* have low-methoxy pectin with a methoxyl content of 6.840%, 4.460%, and 2.348%, respectively [31]. Azad (2014) suggested that lemon pomace has low-methoxy pectin with a methoxyl content ranging between 10.25 ± 0.50 and 4.26 ± 0.01% [39]. Ahmed and their team suggested that ginger lemon, cardamom lemon, and China lemon have low-methoxy pectin with a methoxyl content of 3.97 ± 0.56%, 9.35 ± 0.39%, and *2.86* ± 0.29%, respectively [32]. Dananjaya and their group suggested that *Citrus maxima* have low-methoxy pectin with a methoxyl content of 7.82% [25]. A team led by Norziah suggested that pomelo fruit peels have low-methoxy pectin with a methoxyl content of 5.3% [28].

### 2.4. Anhydrouronic Acid Content

The anhydrouronic acid content of pectin isolated from *Citrus limon, Citrus limetta, Citrus maxima, Citrus sinensis, Citrus jambhiri, Citrus sudachi,* and *Citrus hystrix* was 83.16%, 72.68%, 93.28%, 75.68%, 73.92%, 93.28%, and 91.52%, respectively. Among the three varieties of citrus fruits, the highest AUA content was observed in *Citrus maxima* and *Citrus sudachi,* whereas the lowest AUA content was observed in *Citrus limon* (Figure 5). The content of anhydrouronic acid (AUA) determines the purity of the isolated pectin. It is recommended that the AUA content in pectin should not be less than 65% for use in food additives and pharmaceutical products. The pectin samples isolated from all the citrus fruits had an AUA content greater than 65%; hence, they can be utilized in food and pharmaceutical products.

In the current research work, the anhydrouronic acid content of all the pectin samples was in the range from 72.6% to 93.28%, which is above 65% as recommended by FAO (JECFA 2016) standards, which suggests that the pectin samples are pure and can be utilized in food, cosmetics, and pharmaceutical industries. The present results are supported by previous studies by Kamal and their group, who observed that the AUA content of pectin isolated from *Citrus sinensis* ranged between 38.47 and 41.30% [26]. A team led by Devi observed that the AUA content of pectin isolated from *Citrus limetta* ranged between 91.52% and 51.04% [30]. Kumar and their team observed that the AUA content of pectin isolated from *Citrus hystrix* was 85.07% [29]. Kanmani (2014) observed that the AUA content of pectin isolated from *Citrus sinensis, Citrus limetta,* and *Citrus limon* was 68.74%, 42.80%, and 39.48%, respectively [31]. Azad (2014) suggested that the AUA content of pectin isolated from lemon pomace ranged between 73.22 ± 3.92 and 71.99 ± 0.44% [39]. Ahmed and their team suggested that the AUA content of pectin isolated from ginger lemon, cardamom lemon, and China lemon was 82.12 ± 2.93%, 85.45 ± 4.12%, and 74 ± 1.39%, respectively [32]. Dananjaya and the team observed that the AUA content of pectin isolated from *Citrus maxima* was 68.27% [25]. Norziah and their team observed that pectin isolated from pomelo fruit peels had an AUA content of 74.9% [28]. Tamaki and Konisha (2008) observed that pectin isolated from the peels of *Citrus tankan* had an AUA content of 80% [23].

### 2.5. Degree of Esterification

The degree of esterification determines the probable commercial use of pectin as a gelling and thickening agent. On the basis of the degree of esterification, pectins are divided into two types: (1) high-methoxyl pectin has a degree of esterification greater than 50% and (2) low-methoxyl pectin has a degree of esterification less than 50%. The high-methoxyl pectin forms gels at a low pH (4.0) or in the presence of a low number of soluble solids such as sucrose (55%). The gels prepared from high-methoxyl pectin can be stabilized by hydrophobic interactions. However, the low-methoxyl pectin forms electrostatically stabilized gel networks with divalent cations. The degree of esterification in pectin isolated from *Citrus limon, Citrus limetta, Citrus sinensis, Citrus maxima, Citrus jambhiri, Citrus sudachi,* and *Citrus hystrix* was 42.95%, 41.25%*,* 41.01%, 36.24%, 40.83%, 41.54%, and 42.34%, respectively (Figure 6). Since all the pectin samples had a degree of esterification of less than 50%, they are considered low-methoxyl pectins. This suggests that they can form gels in the presence of divalent cations such as calcium, Cu^2+^, or Fe^2+^.

The degree of esterification of all the pectin samples was lower than 50%, which suggests that they can form hydrogels only in the presence of divalent ions such as calcium, copper, or iron [40,41]. Similarly, a previous study [26] observed that pectin isolated from *Citrus sinensis* has a degree of esterification in the range of 73.26 to 77.56% [26]. A team led by Devi observed that the degree of esterification in pectin isolation from *Citrus limetta* was in the range of 38.46% to 58.62% [30]. A group led by Kumar observed that the degree of esterification in pectin isolated from *Citrus hystrix* was 11.86% [29]. Kanmani (2014) observed that the degree of esterification in pectin isolated from *Citrus sinensis, Citrus limetta,* and *Citrus limon* was 3.50%, 2.98%, and 1.50%, respectively [31]. Azad (2014) observed that the degree of esterification in pectin isolated from lemon pomace ranged between 33.59 ± 0.17 and 79.51 ± 0.36% [39]. A team led by Ahmed observed that the degree of esterification in pectin isolated from ginger lemon, cardamom lemon, and China lemon was 27.38 ± 0.32%, 62.12 ± 1.46%, and 21.96 ± 0.84%, respectively [32]. Methacanon and their team observed that pectin isolated from pomelo (*Citrus maxima)* had a degree of esterification of 57.87% [24]. A team led by Dananjaya observed that the pectin isolated from *Citrus maxima* had a degree of esterification of 72.56% [25]. A group led by Norziah observed that the pectin isolated from pomelo fruit peels had a degree of esterification of 40.5% [28].

### 2.6. Statistical Analysis

The non-metric multidimensional scaling grouped the pectin samples into four groups on the basis of their physicochemical parameters. Group 1 includes the pectin sample isolated from *Citrus maxima,* which was the most different from all the pectin samples with a low equivalent weight, a high AUA content, a low methoxy content, and a low degree of esterification. Group 2 includes the pectin samples isolated from *Citrus sinensis* and *Citrus limetta,* which had a high equivalent weight, a moderate AUA content, a low methoxy content, and a low degree of esterification. Group 3 includes the pectin samples isolated from *Citrus limon* and *Citrus sudachi,* which had a moderate equivalent weight, a high AUA content, a low methoxy content, and a low degree of esterification. Group 4 includes the pectin samples isolated from *Citrus jambhiri* and *Citrus hystrix,* which had a moderate equivalent weight, a moderate AUA content, a low methoxy content, and a low degree of esterification. The dendrogram generated by the multivariate cluster analysis supported the results of the non-metric multidimensional scaling (Figure 7). 

### 2.7. FTIR Analysis for the Identification of Functional Groups

The FTIR spectra and peaks of each pectin sample isolated from different citrus fruit peels were compared with the FTIR data of pectin given by Minhas and their team [42]. It was observed that there was a remarkable difference in the bond pattern of pectin isolated from the peels of each citrus fruit. Pectin isolated from the peels of *Citrus sudachi* had a different FTIR pattern from all the other pectin samples. It had sharp and broad OH stretching vibrations, symmetrical and asymmetrical C–H stretching vibrations, asymmetric C=C=C stretching vibrations, C=O stretching vibrations, broad OH and NH stretching vibrations, sharp C=C Stretching vibrations, CH_2_ symmetric deformation vibration, CH deformation vibration, CH and CH_2_ wagging vibration, sharp C–O stretching vibrations, CH_3_ rocking vibration, CH and CH_2_ out-of-plane deformation vibration, and C–C skeleton vibrations. The FTIR patterns of pectin isolated from the peels of *Citrus limon*, *Citrus limetta,* and *Citrus sinensis* were similar to each other. They contained sharp and broad OH stretching vibrations, symmetrical C–H stretching vibrations, asymmetric C=C=C stretching vibration, C=O stretching vibrations, broad OH and NH stretching vibrations, narrow C=C stretching vibrations, narrow CH_2_ symmetric deformation vibration, CH deformation vibration, CH and CH_2_ wagging vibration, narrow C–O stretching vibrations and CH_3_ rocking vibration, CH and CH_2_ out-of-plane deformation vibration, and C–C skeleton vibrations. The FTIR spectra of *Citrus maxima*, *Citrus jambhiri*, and *Citrus hystrix* were similar to each other. They contained broad OH stretching vibrations, asymmetrical C–H stretching vibrations, asymmetric C=C=C stretching vibrations, broad OH and NH stretching vibrations, narrow C=C stretching vibrations, narrow CH_2_ symmetric deformation vibration, CH deformation vibration, CH_2_ wagging vibration, narrow C–O stretching vibrations, and C–C skeleton vibrations (Figure 8 and Table 1).

From the FTIR analysis, it was observed that there was a remarkable difference in the bond pattern of pectin isolated from the peels of each citrus fruit. All pectin samples had O–H stretching vibrations, which suggest that the absorption in the O–H region is due to the hydrogen bonding in the galacturonic acid polymer at the inter- and intramolecular levels [43]. In all the pectin samples, there was a remarkable presence of bands in the range between 3000 and 2800 cm^−1^, which refers to C–H stretching vibration that includes stretching and bending vibrations from CH, CH_2_, and CH_3_ [43]. The pectin samples from *Citrus limon, Citrus sudachi*, *Citrus maxima*, *Citrus jambhiri*, and *Citrus hystrix* had bands in the range between 2900 and 2950 cm^−1^, which suggests asymmetrical C–H stretching vibration, whereas *Citrus limetta* and *Citrus sinensis* had bands in the range between 2830 cm^−1^ and 2890 cm^−1^ suggesting symmetrical C–H stretching vibration. The C–H stretching in the range of 2950 and 2750 cm^−1^ represents carbohydrate rings and methyl esters of galacturonic acid [44,45]. In all the pectin samples, the ester carbonyl (C=O) groups and carboxylate ion (COO-) were observed, which have bands in the range of 1760–1745 cm^−1^ and 1640–1620 cm^−1^ [31,45]. The asymmetrical stretching bands of the carboxylate group in the range between 1650 and 1550 cm^−1^ were observed in all the pectin samples [45,46]. The C–O and C–C stretching vibrations of the pyranose ring were observed in all pectins that had strong absorption bands in the range from 1150 to 1010 cm^−1^ [43]. The pectin samples *Citrus limon, Citrus sudachi*, *Citrus limetta*, and *Citrus sinensis* had absorption band values higher than 1741 cm^−1^, which indicates that these pectin samples have a high content of hydrophobic groups. However, *Citrus maxima*, *Citrus hystrix*, and *Citrus jambhiri* had absorption band values lower than 1741 cm^−1^, which indicates that these pectin samples have a low content of hydrophobic groups. The degree of methylation value for all the pectin was low, which was depicted through the ratio of a peak area of 1743 cm^−1^ over a total peak area of 1743 and 1637 cm^−1^ [43,47]. The absorption band at 921 cm^−1^ was observed in all the pectin samples, which indicates the presence of D-glucopyranosyl. The absorption band at 827 cm^−1^ was observed in all the pectin samples, which indicates the presence of α-D-mannopyranose [48,49,50,51,52].

## 3. Conclusions

From the present study, it was concluded that the pectins isolated from the peels of all seven citrus fruits (*Citrus limon*, *Citrus limetta*, *Citrus sinensis*, *Citrus maxima*, *Citrus jambhiri*, *Citrus sudachi,* and *Citrus hystrix*) had free galacturonic acid, methoxyl content, and anhydrouronic acid content as per the recommended standards for use in the food and pharmaceutical industries. The degree of esterification was low in all seven pectins, which suggests that it can form a gel in the presence of divalent cations. Hence, this pectin is a good source for use as a thickening agent and for hydrogel formation. From the quantitative analysis, it was observed that a high pectin content was observed in *Citrus sudachi* and a low pectin content was observed in *Citrus maxima*. From the qualitative analysis of pectin, it was observed that the pectin isolated from all the citrus fruit peels had a methoxy content of less than 50%, which suggests that the pectin has the ability to form hydrogel only in the presence of divalent ions. The pectin isolated from all four plants had an anhydrouronic acid content higher than 65%, which suggests that it can be utilized as a thickening agent and for hydrogel formation in the food and pharmaceutical industries. From the FTIR analysis, it was observed that the pectin purified from *Citrus maxima*, *Citrus jambhiri*, and *Citrus hystrix* had few hydrophobic groups whereas pectin purified from *Citrus limon*, *Citrus limetta*, *Citrus sinensis*, and *Citrus sudachi* had many hydrophobic groups. Hence, hydrophobic pectin is suitable for use in the preparation of hydrogels, nanofibers, food packaging material, polysoaps, drug delivery agents, and microparticulate materials, whereas hydrophilic pectin is suitable only for use as gelling and thickening agents.

## 4. Materials and Methods

### 4.1. Sample Collection 

Common citrus fruits (*Citrus limon*, *Citrus limetta*, and *Citrus sinensis)*, were purchased from the local market. Wild citrus fruits such as *Citrus maxima* were collected from the local area of Gauridad (22°24’10.3″ N 70°45’16.9″ E). *Citrus jambhiri* was purchased from a Kerala store (Rajkot, 22°18’06.3″ N 70°48’05.5″ E). *Citrus sudachi* was collected from Kerala (Mankulam, 10°06’44.2″ N 76°55’50.8″ E). *Citrus hystrix* was collected from a farm that was located at Balambha (22°44’02.7″ N 70°25’19.8″ E).

### 4.2. Pectin Extraction and Purification

The fruit peels were removed and washed with distilled water to remove dirt from the surface. Then, they were cut into small pieces, and the fresh weight of the peels was noted. Further, 500 g of peels were ground in the presence of distilled water in a mechanical grinder to prepare a fluidic slurry. The debris was removed from the slurry by filtering it with a nylon cloth. Finally, pectin was precipitated from the filtrate by mixing it with an equal volume of methanol. The precipitate was allowed to dry for 15 days in methanol, and then, the dry weight of pectin was recorded. The yield of pectin obtained from each fruit peel was calculated by using the following equation given by [53]: (1)$$Pectin\;Yield\;(\%)\;=\;(weight\;of\;Pectin)/[weight\;of\;citrus\;fruit\;peels\;\left(g\right)]\;\times \;100$$

### 4.3. Equivalent Weight of Pectin

The equivalent weight in each isolated pectin sample was determined by following the method described by a team led by Khamsucharit [53]. A pectin sample of 0.5 g was moistened with 2 mL of ethanol and then placed in a 250 mL flask. It was dissolved in 100 mL of carbon dioxide-free water, and then, sodium chloride (1 g) was added to it. To determine the endpoint of the titration, 6 drops of phenol red indicator were added to it. The mixture was homogenized thoroughly to dissolve all the pectin particles. Titration was carried out by using 0.1 N sodium hydroxide until the color of the indicator changed to pink (pH 7.5). The neutralized solution was used for the methoxyl determination. The equivalent weight was calculated using the following equation:(2)$$Equivalent\;weight\;\left(g/mol\right)\;=\;\frac{weight\;of\;sample\;(g)}{mL\;of\;alkali\;\times \;normality\;of\;alkali}\;\times \;1000$$

### 4.4. Methoxyl Content of Pectin

The methoxyl content in each isolated pectin sample was estimated by using the titration method described by a group led by Khamsucharit [53]. To measure the methoxyl content, the neutralized solution obtained after the determination of equivalent weight was mixed with 25 mL of 0.25 N NaOH. The mixture was stirred thoroughly and allowed to stand for 30 min at room temperature in a stoppered flask. Further, 25 mL of 0.25 N HCl was then added to the mixture, and then, titration was carried out slowly by adding 0.1 N NaOH up to the endpoint. The methoxyl content in each pectin sample was calculated using the following equation [53]
(3)$$Methoxy\;content\;\left(\%\right)\;=\;\frac{mL\;of\;alkali\;\times \;Normality\;of\;Alkali\;\times \;31}{Weight\;of\;sample\;(gm)\;\times \;100}\;\times \;100$$

### 4.5. Anhydrouronic Acid Content of Pectin

The total anhydrouronic acid content in each pectin sample was calculated from the titration volumes obtained after the determination of equivalent weight and methoxy content. The AUA content was calculated by using the following equation described by Suhaila and Zahariah (1995) [54] and a team led by Khamsucharit [53]: (4)$$AUA\;\left(\%\right)\;=\;\frac{176\times 0.1z\times 100}{w\times 1000}+\frac{176\times 0.1y\times 100}{w\times 1000}$$
where one molecular unit of AUA (1 unit) = 176 g, z = mL (titer) of sodium hydroxide from equivalent weight determination, y = mL (titer) of sodium hydroxide from methoxyl content determination, w = weight of the sample.

### 4.6. Degree of Esterification of Pectin

The degree of esterification of each pectin sample was calculated using the formula described in previous studies [39,53,55]: (5)$$\mathrm{DE}\;\left(\%\right)=\frac{176\times \mathrm{MeO}}{31\times \mathrm{AUA}\;}\times 100$$
where DE = degree of esterification, 176 = molecular weight of anhydrouronic acid, MeO = methoxyl content (%), 31 = molecular weight of methoxyl group, AUA = anhydrouronic acid content (%).

### 4.7. FTIR Analysis of Pectin Samples

The Fourier-transform infrared (FTIR) spectra of all the pectin samples were recorded on an IR prestige-21 (Shimadzu Europa GmbH, Tokyo, Japan). For sample analysis, solid KBr pellets of samples were prepared under 150 kg/cm^2^ hydraulic pressure. The FTIR measurement was taken for all the samples in the range of 400–4000 cm^−1^ at a resolution of 1 nm. 

### 4.8. Statistical Analysis

Non-metric multidimensional scaling was used to group the pectin samples on the basis of their biochemical parameters. Further, multivariate cluster analysis was performed to construct a dendrogram based on the biochemical parameters of pectin using the similarity matrix of the paired group (UPGMA) method with arithmetic averages and the Bray–Curtis similarity index. These analyses were performed using the PAST: Paleontological Statistics software package, version 4.05 [56].

## Figures and Tables

**Figure 1 gels-09-00908-f001:**
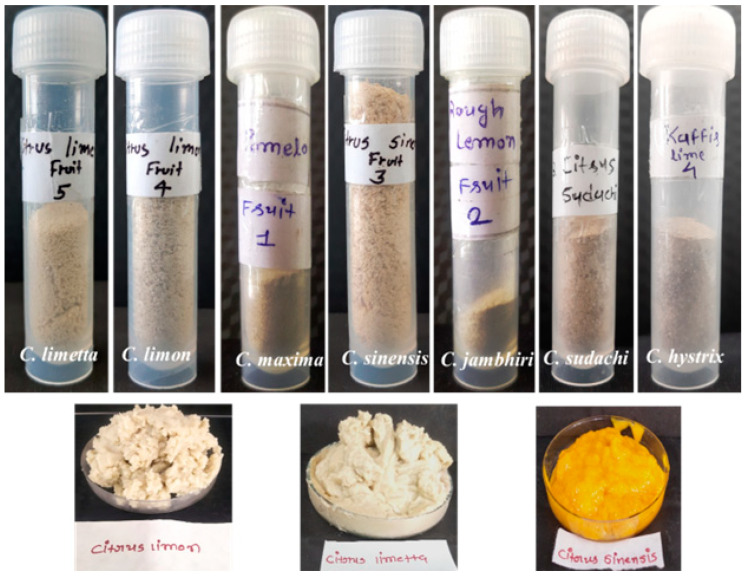
Pectin isolated from different *Citrus* peels.

**Figure 2 gels-09-00908-f002:**
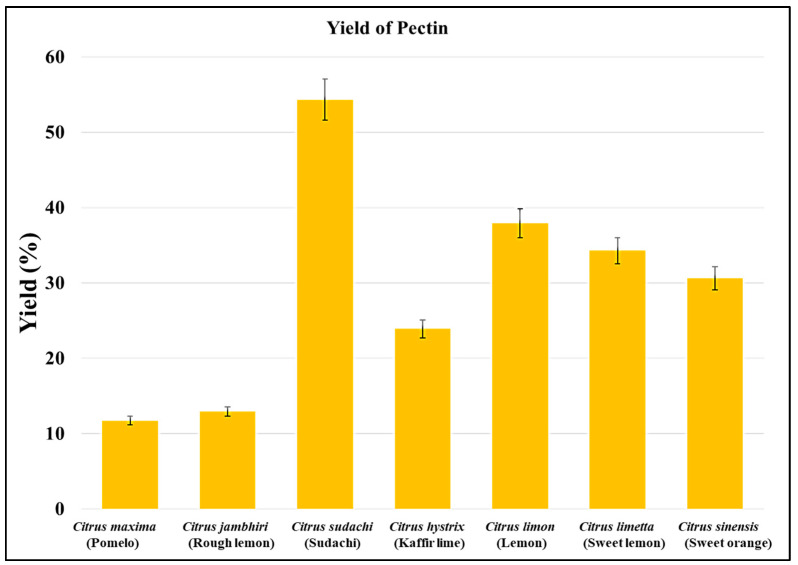
Pectin yields from various citrus peels.

**Figure 3 gels-09-00908-f003:**
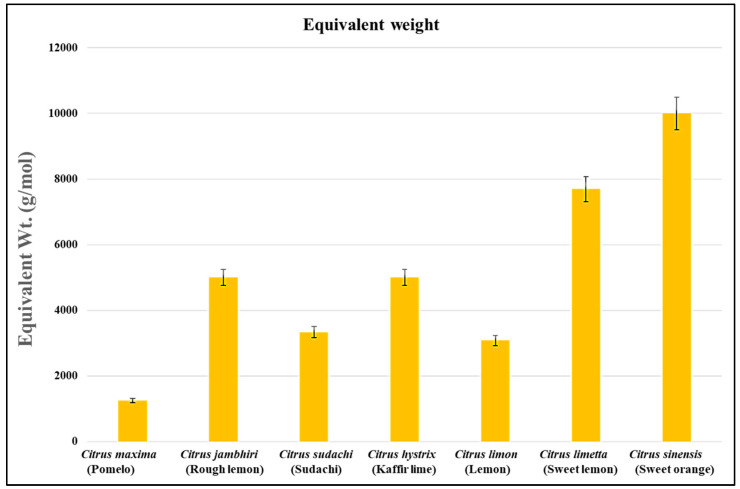
The equivalent weight of all the samples of citrus peels.

**Figure 4 gels-09-00908-f004:**
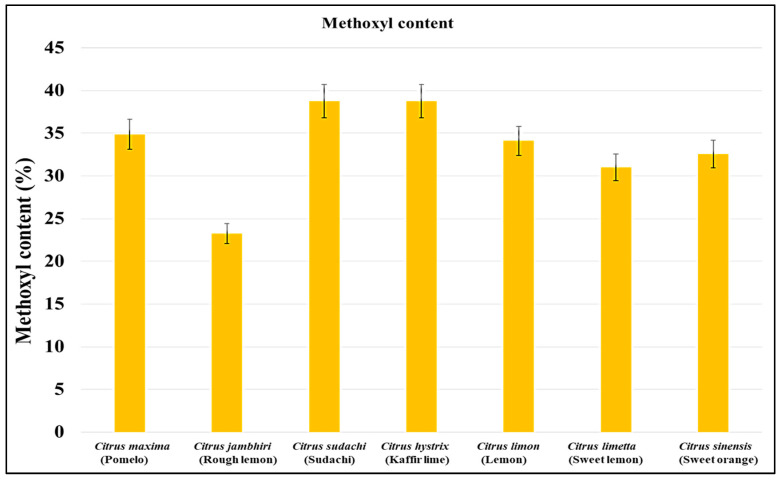
Methoxyl content of all the citrus peel samples.

**Figure 5 gels-09-00908-f005:**
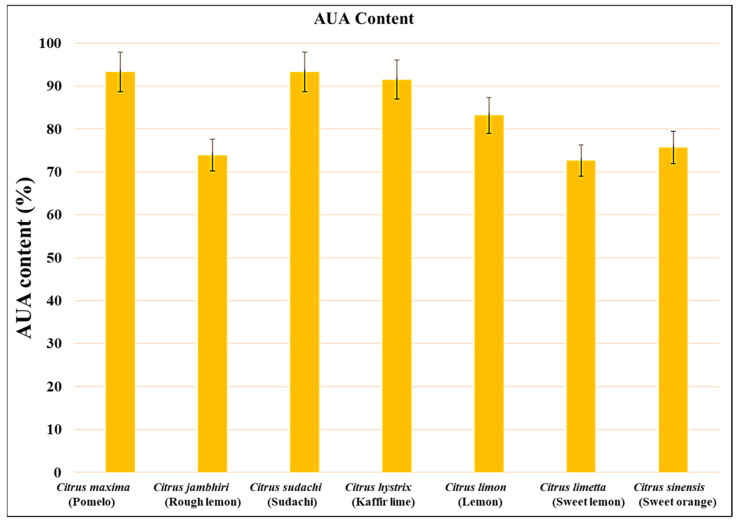
Anhydrouronic acid content of all the citrus peel samples.

**Figure 6 gels-09-00908-f006:**
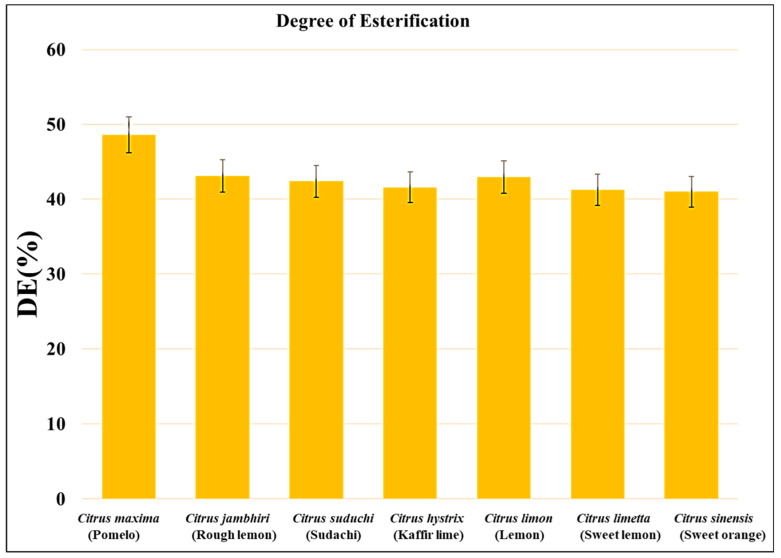
Degree of esterification of all the citrus peel samples.

**Figure 7 gels-09-00908-f007:**
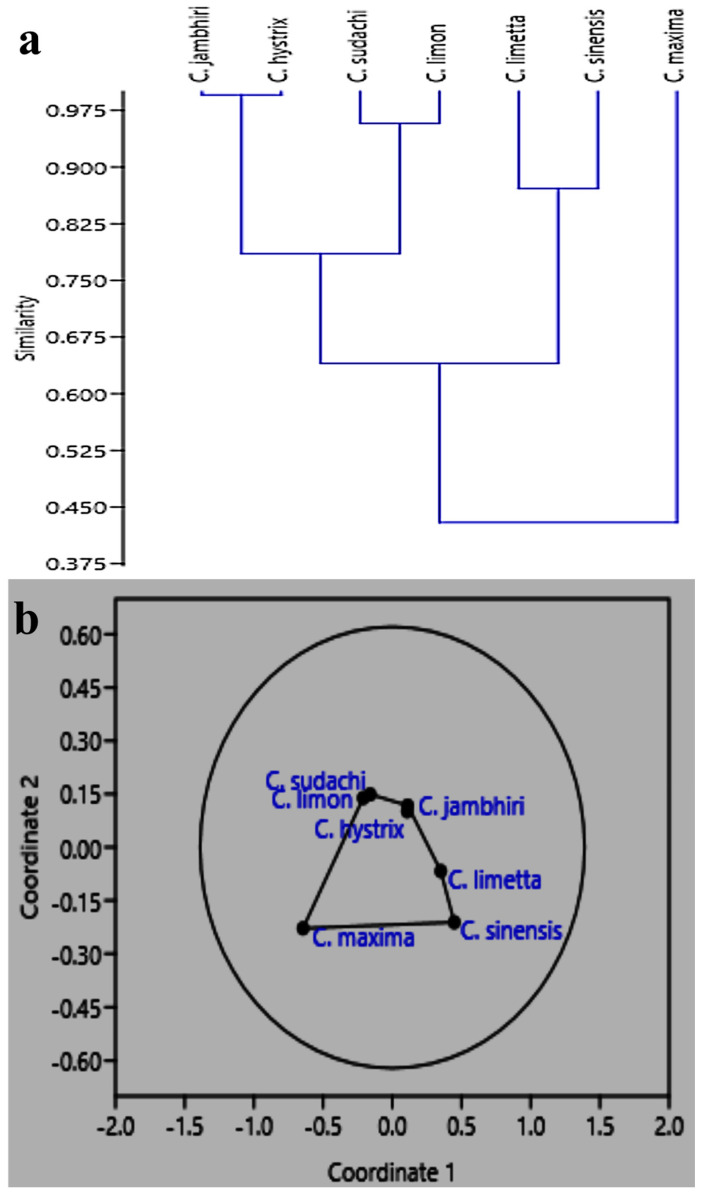
(**a**) Dendrogram based on the paired group (UPGMA) algorithm using the Bray–Curtis similarity index for the clustering of pectin samples using physicochemical parameters. (**b**) Non-multidimensional scaling of pectin samples on the basis of physicochemical parameters.

**Figure 8 gels-09-00908-f008:**
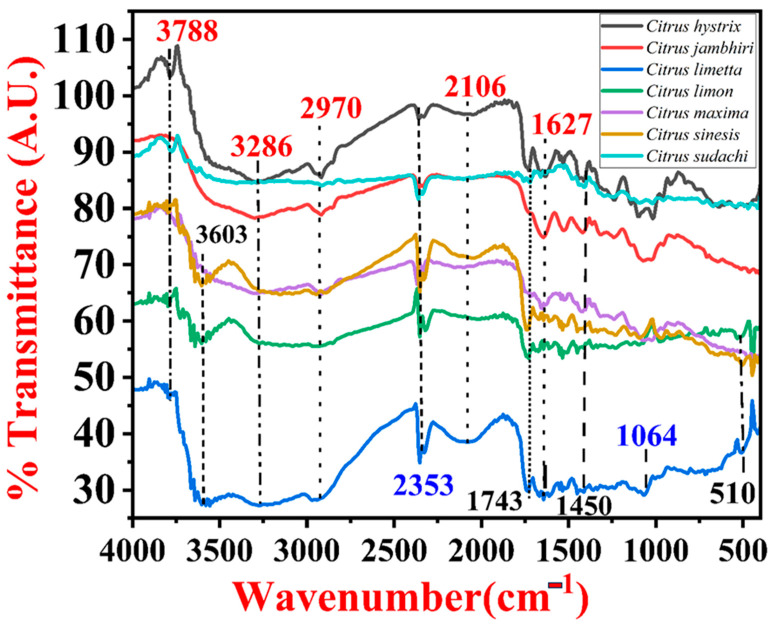
FTIR spectra of pectin isolated from various citrus peels.

**Table 1 gels-09-00908-t001:** FTIR analysis of pectin isolated from different citrus fruit peels.

Sr. No.	Type of Bond	Wave Number (Pectin) from [42]	Wave Number (Pectin) from *C. limon*	Wave Number (Pectin) from *C. sudachi*	Wave Number (Pectin) from *C. limetta*	Wave Number (Pectin) from *C. sinensis*	Wave Number (Pectin) from *C. maxima*	Wave Number (Pectin) from *C. jambhiri*	Wave Number (Pectin) from *C. hystrix*
1.	O–H stretching vibration	3437	3657.16 3309.96	3634.01 3410.26	3649.44 3317.67 3163.36	3618.58 3294.53	3294.53	3302.24	3286.81
2.	C–H stretching vibration- asymmetrical	2931	2947.33	2916.47			2916.47	2924.18	2924.18
3.	C–H stretching vibration- symmetrical	-	2854.74	-	2839.31	2885.60	-	-	-
4.	O–H and N–H stretching vibration	-	2430.39	2360.95	2492.11	2306.94 2492.11	2368.66	2353.23	2353.23
5.	Asymmetric C=C=C stretching vibration		2029.18	2106.34	1921.16 2052.33	2036.90	2106.34	-	2052.33
6.	C=O stretching vibration	1749	1743.71	1720.53	1743.71	1743.71	-	-	1735.99
7.	C=C and COO- Stretching vibration	1628	1527.67	1666.55	1519.96	1604.83 1527.67	1651.22 1527.67	1643.41 1527.67	1635.69 1527.67
8.	CH_2_ symmetric deformation vibration	1444	1435.09	1411.94	1435.09	1435.09	1419.66	1419.66	1411.94
9.	C–H deformation vibration		1350.22	1288.49	1357.93	1350.22	-	1311.64	-
10.	CH_2_ wagging vibration		1226.77	1242.20	1249.91	1234.48	1242.20	1242.20	1234.48
11.	C–O and C–C stretching vibration		1080.17	1087.89	1064.74	1087.89	1072.46	1072.46	1095.60
12.	CH_3_ rocking vibration		972.16	1018.45 956.72	979.87	979.89	-	-	1018.45 964.44
13.	CH_2_ out-of-plane deformation vibration		833.28	-	833.28		895.00	-	825.56
14.	C–H out-of-plane deformation vibration		748.41	732.97	709.83	763.84	-	-	-
15.	C–H wagging vibration		648.10	671.25	617.24	663.53	-	-	640.39
16.	C–C skeleton vibration		516.94 439.78	547.80	486.08	509.22	470.65	-	493.79

## Data Availability

The data presented in this study are available in the article.

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
