# Peer review of "A Comparative Analysis of the Physico-Chemical Properties of Pectin Isolated from the Peels of Seven Different Citrus Fruits"

_gels, 2023, doi:10.3390/gels9110908_

Round 1
Reviewer 1 Report
Comments and Suggestions for Authors
Dear authors,
pectin isolation from fruits has been a subject of many studies, but there is still room for further investigation, as it was proved with your manuscript. The peel of citrus fruits represents one of the great wastes in food industry, and it should be investigated how to use it. Your research gives an insight on the use of seven different citrus fruits for pectin isolation, the properties of isolated pectin and the comparative analysis among seven fruits.
Great work was done and it should be properly written and explained. I have several recommendations for improving your manuscript.
The introduction provides main information required to understand the manuscript. However, it could be improved by explaining the stretching vibration and deformations for better understanding the FTIR results. A picture of pectin structure is optional.
Materials and methods are well described, but please make the mass unit sign (gram) unique - is it gm or g? I recommend to use SI unit g for gram. Check through text.
Please check the spacing through text...there are several extra spacing between words at every page.
I recommend to write '%' sign without spacing between number... or just make it unique through text.
If you write names of fruits in Italic, please make sure that every time it is in Italic (check line 155, 350, 351 and through the text).
Line 315 - please delete "Click or tap here to enter text" and check the sentence.
The results are written very rough, indicating only what is very easy to see on figures or in tables. The discussion section is only a list of previous studies on same or similar fruits and obtained results. There is very poor explanations of, for example, is the different structure of pectin related to the isolation quantity? What to do to improve the isolation? Where to use pectins with different structure and quality? Or similar explanations that prove that this research has a point and meaning in this field of study, not just comparison with previous studies.
After some improvement, this manuscript could be considered for publishing in this journal.
Best regards
Author Response
Author’s Response
|
|
The authors express their gratitude to the respected reviewer for their thoughtful comments and outstanding ideas, which have significantly contributed to the enhancement of our manuscript. The revisions made to the sentences in the manuscript are indicated by highlighting in yellow color
|
|
1 |
The introduction provides main information required to understand the manuscript. However, it could be improved by explaining the stretching vibration and deformations for better understanding the FTIR results. A picture of pectin structure is optional. Author’s Response: Corrections are done as per suggestions of respected reviewer. |
|
2 |
Materials and methods are well described, but please make the mass unit sign (gram) unique - is it gm or g? I recommend to use SI unit g for gram. Check through text. Author’s Response: Corrections are done as per suggestions of respected reviewer. |
|
3 |
Please check the spacing through text...there are several extra spacing between words at every page. Author’s Response: Corrections are done as per suggestions of respected reviewer. |
|
4 |
I recommend to write '%' sign without spacing between number... or just make it unique through text. Author’s Response: Corrections are done as per suggestions of respected reviewer. |
|
5 |
If you write names of fruits in Italic, please make sure that every time it is in Italic (check line 155, 350, 351 and through the text). Author’s Response: Corrections are done as per suggestions of respected reviewer. |
|
6 |
Line 315 - please delete "Click or tap here to enter text" and check the sentence. Author’s Response: Corrections are done as per suggestions of respected reviewer. |
|
7 |
The results are written very rough, indicating only what is very easy to see on figures or in tables. The discussion section is only a list of previous studies on same or similar fruits and obtained results. There is very poor explanations of, for example, is the different structure of pectin related to the isolation quantity? What to do to improve the isolation? Where to use pectins with different structure and quality? Or similar explanations that prove that this research has a point and meaning in this field of study, not just comparison with previous studies. Author’s Response: The difference in FTIR band absorption pattern is compared and on the basis of that difference in hydrophobic and hydrophilic groups present in pectin samples is identified. Based on hydrophilic and hydrophobic nature of pectin samples its application in food, pharmaceutical and food packaging industries is explained in the revised version of the manuscript as suggested by the respected reviewer. |

Reviewer 2 Report
Comments and Suggestions for Authors
In the present manuscript, native pectin was obtained from peels of seven citrus fruits (Citrus limon, Citrus limetta, Citrus sinensis, Citrus maxima, Citrus jambhiri, Citrus sudachi, and Citrus hystrix). The quality of each pectin sample was compared by using parameters such as equivalent weight, anhydrouronic acid content, methoxy content, and degree of esterification. Pectin samples have high content of non-esterified galacturonic acid in the molecular chains which provides viscosity and binding property with water. The methoxy content and degree of esterification of all pectins was lower than 50% which suggests that it cannot easily disperse in water and can form gel only in presence of divalent cations. The anhydrouronic acid content of all isolated pectins samples was above 65% which suggests that the pectin was pure and can be utilized as food ingredient in domestic foods and food industries. From the FTIR analysis of pectin it was observed that the bond pattern of Citrus maxima, Citrus jambhiri, and Citrus hystrix was similar. The bond pattern of Citrus limon, Citrus limetta, and Citrus sinensis was similar. However, the bond pattern of Citrus sudachi was different from all other citrus fruit that it cannot easily disperse in water and can form gel only in presence of divalent cations.
1. I suggest to introduce the meaning of 176 and 31 values in Eq 2-5.
2. The equivalent weight is expressed in mg/mol not gm/mol (line 100, page 3).
3. Please, increase resolution for Figs 2 and 3.
4. Please introduce only one FTIR spectrum (as an example) in Fig 8 because the existence of seven FTIR spectra in Fig 7 makes it difficult their visualization. Table 1 already contains the wave numbers for all types of bonds from seven crude pectins.
The conclusions are consistent with the evidence and arguments presented.
The text is clear and easy to read.
.
Author Response
Author’s Response
|
|
The authors express their gratitude to the respected reviewer for their thoughtful comments and outstanding ideas, which have significantly contributed to the enhancement of our manuscript. The revisions made to the sentences in the manuscript are indicated by highlighting in turquoise color |
|
1 |
I suggest to introduce the meaning of 176 and 31 values in Eq 2-5. Author’s Response: Corrections are done as per suggestions of respected reviewer. |
|
2 |
The equivalent weight is expressed in mg/mol not gm/mol (line 100, page 3). Author’s Response: Corrections are done as per suggestions of respected reviewer. |
|
3 |
Please, increase resolution for Figs 2 and 3. Author’s Response: Corrections are done as per suggestions of respected reviewer. |
|
4 |
Please introduce only one FTIR spectrum (as an example) in Fig 8 because the existence of seven FTIR spectra in Fig 7 makes it difficult their visualization. Table 1 already contains the wave numbers for all types of bonds from seven crude pectins. Author’s Response: Corrections are done as per suggestions of respected reviewer. |

Round 2
Reviewer 1 Report
Comments and Suggestions for Authors
The authors have revised the manuscript according to my comments and therefore I recommend it for publishing.
Good luck!